# Identification and Expression Profile of *CLE41/44-PXY-WOX* Genes in Adult Trees *Pinus sylvestris* L. Trunk Tissues during Cambial Activity

**DOI:** 10.3390/plants12040835

**Published:** 2023-02-13

**Authors:** Natalia A. Galibina, Yulia L. Moshchenskaya, Tatiana V. Tarelkina, Kseniya M. Nikerova, Maxim A. Korzhenevskii, Aleksandra A. Serkova, Nikita V. Afoshin, Ludmila I. Semenova, Diana S. Ivanova, Elena N. Guljaeva, Olga V. Chirva

**Affiliations:** Forest Research Institute, Karelian Research Centre of the Russian Academy of Sciences, 185910 Petrozavodsk, Russia

**Keywords:** cambial age, annual ring width, xylogenesis, phloem formation, conducting phloem, cambial zone, differentiating xylem, *WOXs*, *CLE41/44-PXY* signaling

## Abstract

WUSCHEL (WUS)-related homeobox (WOX) protein family members play important roles in the maintenance and proliferation of the stem cells in the cambium, the lateral meristem that forms all the wood structural elements. Most studies have examined the function of these genes in angiosperms, and very little was known about coniferous trees. Pine is one of the most critical forest-forming conifers globally, and in this research, we studied the distribution of *WOX4*, *WOX13*, and *WOXG* genes expression in *Pinus sylvestris* L. trunk tissues. Further, we considered the role of TDIF(*CLE41/44*)/TDR(*PXY*) signaling in regulating Scots pine cambial activity. The distribution of *CLE41/44-PXY-WOXs* gene expression in Scots pine trunk tissues was studied: (1) depending on the stage of ontogenesis (the first group of objects); and (2) depending on the stage of cambial growth (the second group of objects). The first group of objects is lingonberry pine forests of different ages (30-, 80-, and 180-year-old stands) in the middle taiga subzone. At the time of selection, all the trees of the studied groups were at the same seasonal stage of development: the formation of late phloem and early xylem was occurring in the trunk. The second group of objects is 40-year-old pine trees that were selected growing in the forest seed orchard. We took the trunk tissue samples on 27 May 2022, 21 June 2022, and 21 July 2022. We have indicated the spatial separation expressed of *PsCLE41/44* and *PsPXY* in pine trunk tissues. *PsCLE41/44* was differentially expressed in Fraction 1, including phloem cells and cambial zone. Maximum expression of the *PsPXY* gene occurred in Fraction 2, including differentiating xylem cells. The maximum expression of the *PsCLE41/44* gene occurred on 27 May, when the number of cells in the cambial zone was the highest, and then it decreased to almost zero. The *PsPXY* gene transcript level increased from May to the end of July. We found that the highest transcript level of the *PsWOX4* gene was during the period of active cell proliferation in the cambial zone, and also in the trees with the cambial age 63 years, which were characterized by the largest number of cell layers in the cambial zone. In this study, we have examined the expression profiles of genes belonging to the ancient clade (*PsWOXG* and *PsWOX13*) in stem tissues in Scots pine for the first time. We found that, in contrast to *PsWOX4* (high expression that was observed during the period of active formation of early tracheids), the expression of genes of the ancient clade of the *WOX* genes was observed during the period of decreased cambial activity in the second half of the growing season. We found that *PsWOX13* expression was shifted to Fraction 1 in most cases and increased from the phloem side, while *PsWOXG* expression was not clearly bound to a certain fraction. Based on the data, the role of the CLE41/44-PXY-WOX signaling module in regulating *P. sylvestris* cambial growth is discussed.

## 1. Introduction

Coniferous forests make up one of the largest terrestrial carbon sinks and play an important role in climate change mitigation. Conifers also have an enormous economic importance, as they are a renewable source of timber, paper pulp, and other non-wood products [1]. It is estimated that 50% of the global timber is supplied by conifers, mainly by the genus Pinus, as they generate higher and faster economic yield than angiosperms [2].

The wood (xylem) formation occurs because of the photoassimilate influx from photosynthetic leaves by phloem. The entire diversity of the xylem and phloem structural elements is generated through the activity of the lateral meristem—cambium. Cambium derivatives, facing the outer trunk layers, are differentiated into the structural elements of the phloem, and one faces inside of the trunk, into the elements of the xylem. In contrast to most plant cells, stem cells within cambium are thin and extremely long, and they divide down their long axis in a highly ordered manner, parallel to the tangential axis of the stem [3]. The *WUSCHEL-RELATED HOMEOBOX4* (*WOX4*) gene is involved in the regulation of cambium cell proliferation; it is the identified target of the TDIF(CLE41/44)-TDR(PXY) signaling pathway. Functions of the *CLE-PXY-WOX* signaling module have been widely described in *Arabidopsis thaliana* [4,5,6,7,8,9,10,11,12]. CLE41 (CLAVATA3/ESR LIKE 41) and PXY (PHLOEM INTERCALATED WITH XYLEM) are a ligand–receptor pair, constituting a multifunctional pathway that regulates vascular cell division, vascular bundle organization, and xylem differentiation [3]. The first CLE peptide was isolated from the *Zinnia elegans* culture as a factor stimulating the proliferation of cambial cells and inhibiting the differentiation of vascular elements, and it was called TDIF (Tracheary Element Differentiation Inhibitory Factor) [4,5,6,7]. TDIF is encoded by the *CLE41* and *CLE44* genes in the *Arabidopsis* genome, which are expressed mainly in the phloem tissue and the neighboring cells [4]. Upon cleavage and modification, TDIF is presumably released into the apoplastic space and diffuses toward the cambial cells, where it is bound by the plasma membrane-associated protein PXY/TDIF RECEPTOR (TDR) [4,5,6,7]. The TDIF signal from phloem plays a crucial role in the maintenance of vascular stem cells by two independent pathways: *WOX4*-independent inhibition of the xylem commitment of vascular stem cells and *WOX4*-dependent enhancement of their proliferation [7]. Thus, the *CLE-PXY-WOX* signaling module is important for cambium growth and development. In most studies, the function of these genes has been examined by studying the angiosperms, and we know very little about other plant species.

The *WOX* gene family can be divided into three clades, according to the time of their appearance during plant evolution: the ancient clade, whose members are present in all plant lineages, from green algae to seed plants; the intermediate clade, present in vascular plants; and the modern or WUS clade, only found in ferns and seed plants [13]. The number of *WOX* genes presented in the genomes of different species increases as the plant body plan becomes more complex. The *Arabidopsis* genome encodes at least 15 WOX proteins: a modern/WUS clade, including WUS and AtWOX1-7; an intermediate clade, including AtWOX8, 9, 11, and 12; and an ancient clade, containing AtWOX10, 13, and 14 [13,14,15,16]. All the *WOX* genes examined indicate very specific expression patterns, both spatially and temporally, which are important for their functions [14]. The *WUSCHEL* (*AtWUS*) has been shown to act in the organizing center of the shoot apical meristem (SAM) to maintain the stem cell population [17,18]. A similar role has been proposed for *AtWOX5* in the root apical meristem (RAM) [19] and for *AtWOX4* in the cambial meristem (CAM) [7,20]. The *AtWOX2*, *AtWOX8*, and *AtWOX9* genes have been implicated in the patterning and morphogenesis of the early embryo [16,21,22], and *AtWOX1*, *AtWOX3*, and *AtWOX6* regulate flower development and/or inflorescence architecture [23]. The *A*. *thaliana* ancient clade genes *AtWOX13* and *AtWOX14* are expressed in most tissues and developmental stages (roots, shoots, and reproductive organs) [24]. *Arabidopsis* plants investigation showed that *WOX14* acted redundantly with *WOX4* in promoting vascular cell division downstream of PXY signaling [3]. Unlike *Arabidopsis*, woody plants are perennial plants, with the annual differentiation of vascular tissues [15]. The investigation of Liu B. et al. demonstrated that no *WOX* genes from *Populus trichocarpa* and *P. tomentosa* were classified together with *AtWOX14* [15]. *P. trichocarpa* and *P. tomentosa* also contained three ancient *WOX* genes: *PtrWOX13a, b*, and *c*; these were very similar in sequence, with *PtrWOX13b* and *PtrWOX13c* being sister pairs [15]. Grape also had three ancient *WOX* genes: *VvWOX13A*, *B,* and *C* [25]. In the WOX family, 14 and 10 genes have been identified and described in *Phalaenopsis equestris* and *Dendrobium catenatum*, respectively. It was shown that the *DcWOX4* gene in *D. catenatum* (an orthologue of the *AtWOX4* gene involved in regulating cambial activity in *A. thaliana* [20] and *Populus* [26]), is highly expressed in the young green root part, with relatively weak expression in the leaves, root, and floral parts (sepal and labellum), while the *PeWOX4* gene was relatively highly expressed in the stem and labellum. The *TaWOX11* and *TaWOX5* genes identified in *Triticum aestivum* L. (phylogenetically grouped with *AtWUS* and *AtWOX4*) were highly expressed at different stages of stem and root development [27]. The *PeWOX13A*, *PeWOX13B*, and *PeWOX13C* genes of *P. equestris* and *DcWOX13* of *D. catenatum*, homologous to the *AtWOX13* gene, which is highly expressed in inflorescences and floral buds, and weakly in fruits and leaves [28], showed high expression in all studied tissues [29]. A similar expression pattern was shown for the *AcoWOX13* gene in pineapple [30]. High expression of *WOX13* genes in various tissues in most of the studied plant species [28,29] suggested their different function during plant development. The information about the function of ancient *WOX* genes in conifers is not available [1]. It was shown that the ancient *WOX* clade includes one member in *Pinus sylvestris* (*PsWOX13*) [31], but two members in *Picea abies* (*PaWOX13* and PaWOXG) [14], and three genes in *P. pinaster* (*PpWOX13, PpWOXA*, and *PpWOXG*) and *P. taeda* (*PtWOX13, PtWOXA*, and *PtWOXG*) [32]. The expression patterns of ancient *WOX* genes are present in most tissues, as shown in three-week-old *P. pinaster* seedlings [32]. *WOXA* and *WOXG* have no orthologs in angiosperms [32]. Because of the key roles that WOX proteins play in stem cell maintenance and lateral organ development, WOX proteins are potential targets for better and faster growth of woody plants [15]. However, we have not found functional studies of these genes in gymnosperm species during the cambial growth period.

Scots pine is one of the most critical forest-forming conifers globally, accounting for approximately 39% of all boreal forests [33]. We obtained data on *WOX4* and the ancient *WOX* genes expression distribution in *P. sylvestris* L. trunk tissues during a period of cambial growth. The phylogenetic analyses have identified three members of the ancient clade: *PsWOX13, PsWOXA*, and *PsWOXG*. The expression patterns for the three *WOX* genes (*PsWOX4, PsWOX13*, and *PsWOXG*) were analyzed in the radial row “conductive phloem/cambial zone-differentiating xylem” in different stages of cambial growth period. Further, we examined the role of TDIF/TDR signaling in regulating cambial activity in Scots pine, depending on the ontogenesis stage. Our study investigated pine forests of different ages, formed naturally, including those on the reserve territory (80- and 180-year-old stands) and, therefore, not exposed to anthropogenic impact over the past 80 years.

## 2. Results

### 2.1. CLE41/44 and PXY Gene Identification in the Scots *Pine* Genome

Five genes encoding proteins homologous to PXY and other receptor kinases (RLK) of subgroup XI were identified in the Scots pine genome (Table 1). Comparative evolutionary analysis of the amino acid sequences showed that the *PSY00019884* gene product (91.2% identical amino acids with spruce TDR) (Figure 1) was the closest homologue of *PXY* in pine.

A search of the Scotch pine genome did not reveal genes that could be attributed to the CLE41/44 group, according to the following criteria: the presence of a CLE41/44-specific motif encoding the regulatory dodecapeptide HEVPSGPNPISN; the presence of a sequence encoding a signal peptide. We can explain it because the GymnoPLAZA database contained the nucleotide sequences of 36,106 Scots pine genes sequenced during the implementation of the international ProCoGen project in 2011–2015, while the complete genome of Scots pine contained about over 50,000 genes [34,35].

The partial nucleotide sequence of the coding part of the *P. sylvestris CLE41/44* gene was determined by Sanger sequencing (Appendix A). The GenBank accession number for the nucleotide sequence is BankIt2658773 Pinus OQ200343. This sequence contained the HEVPSGPNPISN CLE motif (Appendix A), a signal peptide with 0.8567 probability. Comparative analysis of the CLE41/44 amino acid sequences derivated from *P. sylvestris* cloned DNA with *P. taeda* (PTA00040742) and *P. pinaster* (PPI00060734) demonstrated 95 and 96% amino acid identity, respectively.

### 2.2. WUSCHEL-RELATED HOMEOBOX (WOX) Gene Identification in the Scots *Pine* Genome

A *Pinus sylvestris* genome search revealed 16 genes encoding protein sequences homologous to the WUS/WOX of the other gymnosperm species described previously (Table 2). Protein structure analysis showed that all sequences contained homeodomains typical for the Wuschel family (PS50071). Comparative evolutionary analysis showed that the amino acid sequences of gymnosperm WUS/WOS proteins formed three characteristic clades: ancient, modern, and intermediate (Figure 2). It was not possible to identify the *WOX4* gene in the genes deposited in GymnoPLAZA. It can be explained because the GymnoPLAZA database contained the nucleotide sequences of 36,106 Scots pine genes sequenced during the implementation of the international ProCoGen project in 2011–2015, while the complete genome of Scots pine contained about over 50,000 genes [34,35], just like for *CLE41/44* (see Section 2.1). Specific expression characterized this gene in the cambial zone of woody plants. For further analysis, the CDS and WOX4 protein sequences deposited with NCBI [36] were used.

The phylogenetic analysis identified three members of the ancient clade. The *PSY00011033* (*PsWOXA*), *PSY00011870* (*PsWOX13*), and *PSY00021883* (*PsWOXG*) genes encoding the WOXA, WOXG, and WOX13 homologues of other pine species were identified (Figure 2). The proportion of identical amino acids in the proteins of *P. sylvestris* and known proteins encoded by the genes of the ancient clade in other members of *Pinus* (*P. pinaster* and *P. taeda*) was over 87%. For the *PSY00011033* (*WOXA*) gene, we failed to construct gene-specific primers for PCR-RT. Therefore, further study of the expression level was conducted only for the *CCP29681.1* (*PsWOX4*), *PSY00011870* (*PsWOX13*), and *PSY00021883* (*PsWOXG*) genes.

### 2.3. Characteristics of Cambial Activity in Pine Trees of Different Ages Growing in Lingonberry Pine Forests

Cambial activity, including intense cell division, is highly changeable and can be varied throughout the plant’s life cycle [4]. For model trees growing in uneven-aged pine forests, we determined the cambial age (C.A.), i.e., the number of annual rings in a radial row on the tissue sampling site (1.5 m). We determined that the C.A. trees of the same age did not differ practically. There were 25 ± 0.8 annual rings for model trees in a 30-year-old pine forest, 63 ± 5.5 annual rings in a 70–80-year-old pine forest, and 164 ± 3.6 annual rings in a 170–180-year-old pine forest. Further, we will consider all the studied characteristics, depending on C.A.

We took cores from model trees at the beginning of the growing season (May) to determine how different cambial activity could be in trees of different ages. It was difficult to determine the boundaries of annual increments in the non-conductive phloem in pine, so we limited the investigation to xylem increments analysis. Tree groups significantly differed both in the width of xylem increments and in the number of xylem cells formed by the cambium over a 5-year period preceding the study (2016–2020). The cambium of young trees (C.A.—25 years old) most actively produced xylem cells, resulting in wide xylem increment formation (Figure 3). The average width of the xylem increment over 5 years in 70–80- (C.A.—63 years old) and 170–180-year-old (C.A.—164 years old) trees was 1.8 and 2.3 times lower, respectively, compared with 30-year-old trees (Figure 3a). This was associated with a 2.1 times lower number of cells in the wood (Figure 3b).

The data obtained are consistent with the known fact that wood is formed depending on the vascular cambium age [37,38]. The young cambium produces wood, which is characterized by thin walls, low density, and a greater radial increment compared with mature wood [39,40].

### 2.4. Characteristics of Xylogenesis, Phloem Formation in the Current Year, and Cambial Zone in Pine Trees of Different Ages

We detected that the width of current year xylem on 21–22 June 2021 was approximately 50% of the average (for the last 5 years) annual increment width (Figure 3a) in pine plants of different ages. The width of the current year’s xylem (Figure 3a) depended on the number of cells in the radial row of the current year’s xylem (Figure 3b). All trees were at the beginning of xylem formation (Figure 4a–c).

We counted the number of phloem cells formed by the cambium from the beginning of the growing season to the tissue sampling date (21–22 June 2021) in model trees at a height of 1.5 m. We found that early phloem (the part of the conductive phloem, which was formed by the cambium at the beginning of the growing season) was fully formed in all the groups of trees. In the phloem of trees of all age groups, parenchyma cells were noted, and they belonged to the elements of the late phloem in the pine (Figure 4a–c). The number of phloem cells formed did not differ between the groups of trees of different ages (Figure 5).

Cells of the cambial zone are organized in radial rows in pine. Each radial row contains one cambial initial and mother cells forming xylem and phloem cells. The cells of the cambial zone differ from the xylem and phloem cells of the extension stage by thinner cell walls and smaller diameters. We counted the number of cells in the cambial zone in pine plants of different ages during the sampling period (21–22 June 2021). We found small, but statistically significant differences in the number of cambial zone cells of trees of different ages (Figure 5). Their number was greater in the 63-year-old cambium.

Thus, based on the characteristics of the conducting tissues and the cambial zone, it can be concluded that the trees of the studied groups were at the same stage of seasonal development at the sampling period for molecular genetic analysis. The formation of late phloem and early xylem occurred during the tissue sampling for the determination of gene expression in the trunks of all tree groups. Fraction 1, prepared from the bark side, included cells of non-conductive phloem, fully formed early phloem, differentiating late phloem, and cambial zone. Fraction 2, prepared from the debarked trunk surface, included differentiating early xylem cells.

### 2.5. Expression of the Genes Encoding CLE41/44-PXY-WOX—Signaling Module in Trunk Tissues of Scots Pine of Different Ages

It was believed that the level of *WOX4* expression correlated strongly with cell proliferation [3,7], and it was used as a genetic marker of this process [41]. We determined that expression of *WOX4* depended on the cambial age. In 30-year-old *P. sylvestris* trees (C.A. 25 years), the maximum expression of *PsWOX4* occurred in Fraction 1 (Figure 5), which, along with phloem cells, included the cambial zone (Figure 4a). With increasing cambial age (trees with C.A. 63, 164 years), the expression of *PsWOX4* shifted to Fraction 2 (Figure 5), which included differentiating early xylem cells (Figure 4b,c). The activity of *WOX4* orthologs in the cambium was found in many plant species, including gymnosperms [42]. Interestingly, expression of *PaWOX4* in *P. abies*, high values of which were detected in the cambium, was not limited to any tissue types, and this expression was detected in all tissues analyzed [14]. The *PsWOX4* transcript level in Fraction 2 (8.75 ± 4.05 arb. Unit) in trees with the cambial age 63 years was higher compared to trees with C.A. 25, 164 years (0.77 ± 0.15 and 1.42 ± 0.68 arb. Unit, respectively). The number of cambial zone cells was also higher (7.2 ± 0.1 in trees with C.A. 63 years versus 6.2 ± 0.2 and 6.4 ± 0.2 in trees with C.A. 25, 164 years) (Figure 5).

It is known that the ancient *WOXs* genes are expressed in various tissues and organs, both in angiosperms and in gymnosperms [14,15,24,25,31,32]. In this study, we attempted to study the profiles of their expression in trunk tissues in pine plants of different ages. We did not find any patterns in the distribution of the transcripts of ancient *WOXs* genes in the radial row “conductive phloem/cambial zone—differentiating xylem” in plants of different ages. We found the patterns of *PsWOX13* expression in Fraction 1 and 2 in all subjects studied. Only in trees with cambial age 164 years was the *PsWOX13* transcript level higher in Fraction 1 compared to Fraction 2 (Figure 5). We indicated the maximum expression of PsWOX13 in trees with cambial age 63 years, in Fraction 1 (Figure 5). The expression level of *PsWOXG* was higher in trunk tissues selected from the xylem side (Fraction 2) in older trees (C.A.—63 and 164 years old), and, on the contrary, in young trees (C.A.—25 years old) in Fraction 1. Further, the maximum *PsWOXG* transcript level was shown in trees with a cambial age of 63 years (Figure 5).

*WOX4* is the identified target of the *CLE41/PXY* signaling pathway, which is an evolutionarily conserved program for regulating vascular cambium activity between angiosperm and gymnosperm tree species [26]. We showed the spatial separation of *PsCLE41/44* and *PsPXY* expressed in pine trunk tissues. *CLE41/44* was differentially expressed in Fraction 1, including phloem cells and the cambial zone. The maximum expression of the *PXY* gene occurred in Fraction 2, including differentiating xylem cells. Interestingly, the *CLE41/44-PXY* transcript level in the trunk tissue of *P. sylvestris* did not depend on cambial age (Figure 5).

We hypothesized that at the time of tissue sampling (21–22 June), the influence of the *CLE41/PXY* signaling pathway on the proliferation-differentiation of cambial initials was reduced. So, in 2022, we studied changes in the expression of *CLE41/PXY/WOXs* genes in trunk tissues of 40-year-old pine trees at different stages of cambial growth.

### 2.6. Characteristics of Cambial Activity in the 40-Year-Old Pine Trees at Different Stages of Cambial Growth

The study of the dynamics of the cambial activity of the trees growing in the seed orchard showed that in 2022, the division of cambial cells began earlier than 5 May. We observed the deposition of the first xylem elements by the cambium on 16–20 May (data not showed). On the first sampling date (27 May), early phloem was formed in all the trees and early xylem was formed. On the second sampling date (21 June), the cambium formed cells of the late phloem, while the formation of early tracheids continued from the xylem side. By the third sampling date (21 July), cambial activity decreased, the late phloem was almost completely formed, and late tracheids were formed from the xylem side (Appendix A).

The C.A. of the tissue sampling site (1.5 m) was 34 ± 0.9 annual rings for trees from the 40-year-old pine forest, growing in a forest seed orchard. The width of the annual ring and the number of cells in the radial row of the wood (the average meaning for the last 5 years, 2017–2021) were 2.09 ± 0.23 mm and 60 ± 7, respectively (Figure 6). The width and the number of cells in the radial row of the current year’s xylem were 6%, 40%, and 65% from the average meanings for five years at the time of tissue selection (27 May, 21 June and 21 July, respectively) (Figure 6). The number of cells in the radial row of the conducting phloem increased from 3.1 ± 0.3 (27 May) to 9.5 ± 0.9 (21 July) (Figure 7). The number of cells in the radial row of the cambial zone was the largest on 27 May (15.2 ± 1.3) and decreased to 9.6 ± 1.0 and 9.0 ± 0.6 (21 June and 21 July, respectively) (Figure 7).

On 23 May, Fraction 1 included the cambial zone, early conducting phloem, and a small portion of non-conducting phloem (Appendix A). On 21 June, Fraction 1 contained the cambial zone, the early conductive phloem, a small portion of the non-conductive phloem, and 1–2 cells of differentiating xylem closest to the cambium (Appendix A). On 21 July, Fraction 1 included five tissues: 1–2 cells of the differentiating xylem closest to the cambium, the cambial zone, early conducting phloem, differentiating late conducting phloem, and a small portion of non-conducting phloem (Appendix A). Fraction 2 included a differentiating xylem at all periods of sampling and contained only expanding xylem cells (Xyl (e)) (23 May) or expanding xylem cells and 3–5 xylem cells, which were forming secondary cell walls (21 June and 21 July) (Appendix A).

### 2.7. Expression of the Genes Encoding CLE41/44-PXY-WOX Signaling Module in the 40-Year-Old Pine Trees at Different Stages of Cambial Growth

Maximum expression of the *PsCLE41/44* gene occurred in Fraction 1 on 27 May, when the number of cells in the cambial zone was the highest, and then the *CLE41/44-PXY* transcript level decreased to almost zero (21 July). In Fraction 2, expression of the *PsCLE41/44* gene was practically undetected, except for the selection on 21 June, when the *CLE41/44-PXY* transcript level was higher than in Fraction 1 on 27 May (Figure 7).

*PsPXY* expression was evenly distributed between Fraction 1 and Fraction 2. By the second and third sampling dates (21 June and 21 July), it increased, with a predominance in Fraction 2 (Figure 7).

We found the maximum expression of the *PsWOX4* gene only on 27 May in Fraction 2. On the contrary, the ancient *WOXs* genes were expressed in Fraction 1. The maximum expression of the *PsWOXG* gene occurred only on 21 of June, while the transcript level of the *PsWOX13* gene increased significantly until 21 July (Figure 7).

We could not compare the distribution of the expression of the ancient *WOXs* genes in the cambial zone of 40-year-old pine trees with that of pine trees of different cambial ages (trees with C.A. 63, 164 years). We assume that the reason for this was the different weather in the growing season of 2021 and 2022. Daily average temperatures in May–June 2021 exceeded those in 2022 (Appendix A).

## 3. Discussion

Wood is the most important renewable energy source, and it is also becoming increasingly important as an industrial raw material for the production of numerous products. On a planetary scale, the formation of wood, namely, the capture and long-term storage of carbon dioxide, is of great importance for climate regulation. Despite over 20 years of research history, the mechanisms underlying the regulation of xylogenesis are still not fully understood. The CLE41-PXY-WOX4 signaling module, which is of great importance for cell division in the cambial zone and the formation of xylem, has been studied in *Arabidopsis* plants [5,7,9,43,44,45,46,47]. *P. trichocarpa* is considered a model species for studying the molecular genetic aspects of xylogenesis regulation among woody plants [48,49,50,51]. Usually, studies are conducted on seedlings, in which one or two rings of juvenile wood are formed. Published transcriptomes of woody plants contain information on the distribution of *CLE41-PXY-WOX4* gene expression in trunk tissues of adult plants (see, for example [41]), but such works are unitary and do not contain information about the seasonal dynamics.

WOX4 plays a major role in controlling cell identity and division activity in the vascular cambium, as shown in hybrid aspen [26]. Downregulation of *WOX4* homologs by RNA interference in hybrid aspen causes more dramatic phenotypic changes than are observed in annual species; in the most extreme cases, the resulting reductions in cambial activity and wood formation are severe enough to prevent the trees from remaining upright [4]. We found in 40-year-old pine trees that the highest transcript level of the *WOX4* gene in Fraction 2 was observed during the period of active cell proliferation in the cambial zone (27 May 2022). Further, the trees with the cambial age 63 years, comparable to other trees (C.A. 25 and 164 years), were characterized by: (1) a greater *PsWOX4* transcript level in Fraction 2 and (2) the largest number of cell layers in the cambial zone. In studies performed on *Arabidopsis*, it was shown that the maximum expression of *WOX4* was observed in part of the xylem-facing cambium [52,53]. We had shown that *PsWOX4* gene expression patterns (trees with C.A. 25, 63, and 164 years) were observed both in Fraction 2 and Fraction 1. Our data confirmed previous studies on woody plants. So, expression of *PaWOX4* in *P. abies*, high values of which were detected in the cambium, was not limited to any tissue types, and this expression was detected in all tissues analyzed [14]. Previously, we found that the expression level of *BpWOX4* was higher in the cambial zone during the period of the highest cambial activity (11 June), and it was higher in the differentiating xylem during the period of decreased cambial activity (25 June) [54]. It has been indicated many times in the literature that the level of *WOX4* expression correlated strongly with cell proliferation [3,7,55] and was used as a genetic marker of this process [41]. We suggested that during the period of cambial growth, active cell proliferation predominated in the xylem part of the cambium, but could shift towards the phloem part of the cambium (in our work trees with C.A. 25 years) under changing conditions.

The activity of *WOX4* genes involved positive regulation through the TDIF/TDR signaling pathway (Figure 8). Interactions between the peptide ligand TDIF/CLE41 and the TDR/PXY receptor have three independent effects on vascular development-related processes: they (1) promote cambial cell proliferation in the procambium/cambium; (2) inhibit xylem cell differentiation; and (3) control vascular patterning [4,5,6,7,8,9,10,11,12]. We have indicated the spatial separation expressed in *PsCLE41/44* and *PsPXY* in pine trunk tissues. *CLE41/44* differentially expressed in Fraction 1, including phloem cells and cambial zone. Maximum expression of the *PXY* gene occurred in Fraction 2, including differentiating xylem cells. Previously, it has been indicated many times in the literature that the spatial separation of the ligand (TDIF) and receptor (TDR) was necessary for the correct spatial orientation of the vascular pattern and the restriction of the stem cell zone [56]. *CLE41/44* expression found in the phloem did not overlap with *TDR/PXY* expression, which was specific to cambial cells [5,6,7,10,44].

Interestingly, the *CLE41/44-PXY* transcript level in trunk tissues of *P. sylvestris* did not depend on cambial age. Previously, we have shown that the expression maximum of *CLE41/44* and *PXY* genes were separated not only in space (conductive phloem/cambial zone—differentiating xylem), but also in time. We found that the highest expression of the genes *BpCLE41/44* was observed on 28 May and preceded the expression maximum of the *BpPXY* gene and the active cambial growth in *B. pendula* [54]. The data obtained on 40-year-old pine trees also showed that the highest *PsCLE41/44* expression in Fraction 1 was observed on 27 May 2022. We found that on 21 June 2022, the transcript level of the *PsCLE41/44* gene in Fraction 2 was higher than in Fraction 1 on 40-year-old pine trees. Thus, a decrease in the number of dividing cells in the cambial zone was accompanied by a decrease in *CLE41/44* expression on the phloem part. As the late phloem develops, *CLE41/44* expression can remain high in xylem cambium derivatives. It is well known that the phloem peptide signal acted as a positional signal for the orientation of periclinal divisions of cambial cells. Increased expression of *CLE41* using the *35S* promoter (*35S::CLE41*, ubiquitous expression of *CLE41*) or *IRX3* promoter (*IRX3::CLE41*, expression of *CLE41* from the xylem side) led to a change in the orientation of cell divisions in the cambial zone, as shown in *Arabidopsis* plants [46,57]. There are several types of cell division in the cambial zone. During periclinal divisions, the cell plate is laid perpendicular to the trunk radius; the result of such divisions is the formation of rows of phloem and xylem derivatives. Anticlinal divisions lead to an increase in the number of cambial cells and, as a result, an increase in the number of rows of xylem and phloem cells. Pseudotransverse divisions of fusiform cambial initials result in the formation of axial parenchyma strands in phloem or xylem. Periclinal cell divisions dominate the active cambium; anticlinal and pseudotransverse divisions account for less than 10% of all mitoses [58,59]. At the same time, the maximum frequency of periclinal divisions in trees of the boreal zone is observed in the first half of the growing season (mid-May to mid-June) [60,61,62,63]. In the second half of June, the frequency of anticlinal and pseudotransverse divisions increases, and their maximum in pine occurs in July [60,61]. Anticlinal and pseudotransverse divisions occur more frequently in cambial initials and phloem mother cells; in xylem mother cells, these types of divisions were observed singly [10,64,65,66,67]. It is possible that the presence of *CLE41/44* gene transcripts in the second half of June–early July (depending on weather conditions) in xylem mother cells may be a common phenomenon for woody plants.

In contrast to the *PsCLE41/44* gene, *PsPXY* expression: (1) was present in Fraction 1 and 2, with a predominance in the latter one; and (2) and was increased by 21 July 2022, as shown in 40-year-old pine trees. It was known that *WOX4* integrates TDIF(CLE41)/TDR(PXY) signaling and auxin signaling for cambium division [7,11,68,69,70] (Figure 8). If TDIF/TDR signaling stimulates *WOX4* transcription and promotes cambium proliferation in stems [7], then auxin signaling attenuates the activity of the stem cell-promoting *WOX4* gene, and the cell autonomously restricts the number of stem cells in stems [68]. ARF5 (auxin response factor) protein limits the number of stem cells through the weakening of *WOX4* activity and increases the expression of the *HB8* gene, positively affecting xylem differentiation [4,71]. Some suggest that ARF5 and PXY form a negative feedback loop. ARF5 activates *PXY* expression and PXY interaction with CLE41/44 peptide. TDIF(CLE41)/TDR(PXY) signaling (1) promotes initial cambial cell divisions and (2) also suppresses ARF5 activity [72,73]. Therefore, high *PsPXY* expression (in the absence of CLE41/44) may indicate an increase in auxin signaling on 21 July 2022. We had previously shown on different forms of silver birch that the maximum expression of the gene *BpWOX4* was due to continuing TDIF/TDR signaling and reduced auxin signaling [54,74].

In *Arabidopsis*, *AtWOX14* has been suggested to act redundantly with *AtWOX4* to control vascular cell proliferation [3]. Both *AtWOX4* and *AtWOX14* accumulate in cambial cells; however, the accumulation of *AtWOX14* in the stem base suggests *AtWOX14* is more important during secondary growth than during primary vascular bundles development [75]. Interestingly, the *WOX14* gene belongs to the ancient clade, but it is found only in *Brassicaceae*. In this study, we examined the expression profiles of genes belonging to the ancient clade in stem tissues in Scots pine for the first time. We found that *PsWOX13* expression was in Fraction 1 in most cases, and it increased from the phloem side until the end of July inclusively. It was believed that representatives of the ancient clade were expressed in different organs (roots, shoots, and reproductive organs) and at different developmental stages [24]. Despite the ubiquitous expression of *WOX13*, the expression pattern was more confined to certain tissues in some plants. In *Panax ginseng*, *PgWOX13a* was expressed in xylem rays, parenchyma, and cambium of the main root, while *PgWOX13b* was expressed in the cortex [35]. Previous studies on *Populus* showed the role of *WOX13* genes during regulation of cambial activity or later stages of wood formation [55]. Interestingly, the *WOX13* genes in *Gossypium hirsutum* (*GhWOX13*) were highly expressed in the cotton fiber, and their expressions gradually increased during the fiber elongation of cotton fibers. In addition, all the *GhWOX13* genes had putative GA, NAA, and BR response elements in their promoter regions and could be induced by these hormones [76]. These results showed that *GhWOX13* genes may have played an important role in cotton fiber development. Although the gymnosperm and angiosperm species, which have a common ancestor ca 300 million years ago, share many morphological and physiological features, it is believed that there are key differences, such as the patterning during the differentiation of cambial derivatives [27]. There are practically no data in the literature on the distribution of *WOX13* expression in the trunk tissues of the conifers. Ancient-clade genes are constitutively expressed in all developmental stages of SE, but also in all plantlet tissues analyzed in *Picea abies* and *Pinus pinaster* [14,27], which is consistent with what was previously reported in angiosperms, although their function in conifers still remains unknown [1]. For example, in *P*. *abies*, expression of *PaWOX13* was detected in both during embryogenesis and in more adult tissues [14]. Similar results were obtained for *P. pinaster*—they observed the expression patterns of *PpWOX13* during embryo development and germination [27]. There were more transcripts of *PpWOX13* in the shoot apex and root tip, compared with hypocotyl and cotyledon in different parts of the three-week-old seedlings of *P. pinaster* [27]. Further, in one-month-old seedlings of *P. pinaster* the high expression of *PaWOX13* was detected in cotyledon vascular, root vascular, and root meristem [27]. Based on our results and literature data, there is an assumption about the role of *PsWOX13* during the period of decrease in cambial activity in the second half of the growing season. Confirmation of this hypothesis requires further research. The ancient clade members WOXA and WOXG did not have angiosperm orthologues. Transcripts of the *PpWOXA* and *PpWOXG* were detected in all the developmental stages during somatic embryo development, and in different germination stages and tissues in seedlings from zygotic embryos, as show on *P. pinaster* [27]. Further, the expression patterns of *PpWOXG* were higher during embryo development than during germination [27]. Our results showed that the expression of *PsWOXG*, unlike *PsWOX13*, was not bound to any fraction and was observed during the period of cambial growth in all studied pine trunk tissues.

## 4. Conclusions

The study of the regulatory mechanisms of cambial growth has been carried out for several decades. Scientists have accumulated a large amount of data on anatomical and biochemical changes in the cells of the cambial zone during the period of activity; many works have been devoted to the hormonal regulation of this process. Elements of the CLE41/44-PXY-WOX signaling module are involved in regulating cambial activity and have been identified in several angiosperm and gymnosperm species of woody plants. However, the expression of the corresponding genes has been studied mainly in young plants less than 10 years old. This study represented a step in our understanding of the role of CLE/PXY/WOX signaling in regulating *P. sylvestris* cambial growth. In our work, we focused on the study of adult trees aged from 30 to 180 years, growing under natural conditions without anthropogenic influence. We also performed a microscopic analysis of developing tissues to accurately characterize the stage of cambial activity. We have shown that during the period of cambial growth in *P. sylvestris*, the expression of the genes’ *CLE-PXY-WOX* patterns were extended in a radial row, “the conductive phloem/the cambial zone (Fraction 1)—the differentiating xylem (Fraction 2)”, with the maximum for each gene in that or another faction. The transcript level of the genes changed during cambial growth (27 May–21 July). In this study, we examined the expression profiles of genes belonging to the ancient clade (*PsWOXG* and *PsWOX13*) in stem tissues in Scots pine for the first time. We believe that further research should focus on studying the dynamics of the level of *CLE41/44-PXY-WOXs* gene transcripts during the growing season, which will make it possible to investigate our hypotheses in the future.

## 5. Materials and Methods

### 5.1. Study Objects

The study occurred in the middle taiga subzone. We selected objects in the same-aged lingonberry pine forests for the study, common in the northern and middle taiga. We established sample plots in May 2021, as we described it earlier [77]. The pine age line was represented by three groups: (1) 30 years (Plot 4)—the beginning of the vegetative–reproductive ontogenesis stage; (2) 70–80 years (Plots 9, 1)—the middle of the vegetative–reproductive ontogenesis stage, as well as the age at which the wood quality was a significant indicator, since pine stands were assigned for felling following forestry requirements in this period; (3) 170–180 years (Plot 3)—the age at the end of the vegetative-reproductive stage of ontogenesis and the beginning of the extinction stage of tree growth (Appendix A). We carried out a continuous count of trees in 2 cm steps of diameter on the sample plots. The following stand characteristics were determined: tree species composition, density, the sum of cross-sectional areas (basal area per hectare), average diameter and height, and growing stock [77]. Three (Plot 9), four (Plot 1), and five (Plots 4, 3) model trees were selected on the sample plot. We chose the dominant trees without suppression signs and damage or damage as model trees. The trunk tissue samples were taken in the cambial growth period (21–22 June 2021).

To study the expression of *CLE41/PXY/WOXs* genes in trunk tissues at different stages of cambial growth, eight 40-year-old pine trees were selected growing in the middle taiga subzone of the Petrozavodsk forest seed orchard of the 1st order (61.91972° N; 34.41389° E). We took the trunk tissue samples on 27 May, 21 June, and 21 July 2022.

### 5.2. Plant Sampling

We took the trunk tissue samples at a height 1.5 m above ground level. For microscopic analysis, blocks (1 × 1 × 0.8 cm, length × width × height) were cut out of the trunk and placed in 3% glutaraldehyde solution. For molecular genetic analysis, “windows” were cut out of the trunk, and we separated the bark from the wood (Appendix A). During the period of cambial growth, the bark moved away from the wood along the expanding xylem zone. Tissue complexes, included cells of non-conductive phloem, fully formed early phloem, differentiating late phloem, and cambial zone (Fraction 1), were prepared from the inner surface of the bark. The layers of tissue, including differentiating early xylem cells (Fraction 2), were scraped off the exposed wood surface with a blade (Appendix A). We monitored the sampling of stem tissues under a light microscope (Figure 4 and Appendix A). The material was frozen in liquid nitrogen and stored at −80 °C.

### 5.3. Microscopy

Three samples (5 × 5 × 3 mm, length × width × height), including the phloem, cambial zone, and last 2–3 annual increments of wood, were cut from every block (1 × 1 × 0.8 cm, length × width × height). We did sample preparation for microscopy, as described previously [54,78]. We made anatomical measurements following guidelines using panoramic cross-sections [79,80,81]. The number of biological replicates (i.e., model trees) was fivefold for the 30-year-old and 170–180-year-old groups, sevenfold for the 70–80-year-old groups, and eightfold for the 40-year-old. The number of technical replicates ranged from three to five for each model tree.

### 5.4. Gene Retrieval from the Scots Pine Genome by Bioinformatics Methods

The search for the *CLE41/44*, *PXY*, and *WOX4* genes was carried out using the *P. sylvestris* gene set deposited in the GymnoPLAZA database (https://bioinformatics.psb.ugent.be/plaza/versions/gymno-plaza/, accessed on 20 July 2022) [82]. To this end, the CDS of *Arabidopsis thaliana, Picea abies*, and several *Pinus* species *CLE41/44*, *PXY*, and *WOX4* genes and the amino acid sequences of corresponding proteins were obtained from The Arabidopsis Information Resource (TAIR) database (https://www.arabidopsis.org, accessed on 20 July 2022), ConGenIE (https://congenie.org/, accessed on 20 July 2022), and the NCBI database (https://www.ncbi.nlm.nih.gov/, accessed on 20 July 2022). We then used the resulting sequences as a BLAST search query across the gene set of *P. sylvestris* to identify homologous sequences.

We predicted the structures of candidate proteins using the National Centre for Biotechnology Information (NCBI) resource (http://www.ncbi.nlm.nih.gov/Structure/cdd/cdd.shtml, accessed on 20 July 2022) [83]. Prediction of protein subcellular localization was performed using DeepLoc 2.0 [84]. Prediction of transmembrane helices was done using DeepTMHMM [85]. We carried out phylogenetic analysis and construction of phylogenetic trees using MEGA X software, as described previously [54]. The percent identity of proteins was determined using the EMBOSS Needle online tool (https://www.ebi.ac.uk/Tools/psa/emboss_needle/, accessed on 20 July 2022).

### 5.5. Sequencing

The RNA sample was isolated from the xylem of the Scotch pine trunk by the CTAB method and used as a template for the reverse transcription (RT) reaction. The RT reaction was carried out using the MMLV RT kit (Evrogen, Moscow, Russia), according to the manufacturer’s protocol. The cDNA obtained during RT was used for PCR.

The potential *CLE41/44* gene and protein PPI00060734 for *P. pinaster* and PTA00040742 for *P. taeda* were found in the GymnoPlaza database (https://bioinformatics.psb.ugent.be/plaza/versions/gymno-plaza/ accessed on 20 July 2022), containing the CLE motif HEVPSGPNPISN. We aligned these nucleotide sequences using the MEGA X software package [86]. Forward 5′-GTATGGCGGATGGTTTTG-3′ and reverse 5′-ATTACTAATTGGATTTGGACCG-3′ primers were designed using Beacon Designer 7 software (PREMIER Biosoft International, USA) for identical regions of these sequences. The primers were synthesized by Syntol (Moscow, Russia). The developed primers were used for PCR and *P. sylvestris CLE41/44* gene sequencing. We performed amplification on a ThermalCycler T100 instrument (Bio-Rad, Hercules, CA, USA) using KAPA3G Plant PCR Kits (Kapa Biosystems Pty (Ltd), Cape Town, South Africa). Amplification reaction (volume of 25 µL) contained 1xKAPPA Plant PCR Buffer, forward and reverse primers, (0.3 µM both), 1U KAPPA3G Plant DNA Polymerase, and cDNA—50 ng. The PCR products were purified using the ExoSAP-IT Express PCR Product CleanupReagent kit (AppliedBiosystems, Thermo Fisher Scientific Baltics UAB, Vilnius, Lithuania) and used as a material for sequencing. Sanger sequencing was performed using the Big DyeTerminatorv3.1 CycleSequencing Kit (Applied Biosystems, Thermo Fisher Scientific Baltics UAB, Vilnius, Lithuania), with the same primers used for amplification. Purification of fluorescent dyes was carried out by the BigDye^®^ Xterminator Purification Kit (AppliedBiosystems, Thermo Fisher Scientific Baltics UAB, Vilnius, Lithuania). Nucleotide sequences were determined on a SeqStudio genetic analyzer (Applied Biosystems, Thermo Fisher Scientific Baltics UAB, Vilnius, Lithuania). The amino acid sequence was translated from the resulting nucleotide sequence using the EMBOSS Sixpack resource https://www.ebi.ac.uk/Tools/st/emboss_sixpack (accessed on 20 July 2022). We checked the translated amino acid sequences for the dodecapeptide characteristic of CLE41/44 (HEVPSGPNPISN). The dodecapeptide-containing sequence was then tested for the presence of the signal peptide using the SignalP 6.0 service (https://services.healthtech.dtu.dk/service.php?SignalP accessed on 20 July 2022). Alignment of the amino acid sequences of CLE41 peptides from *P. pinaster* and *P. taeda* with the sequence derivated from *P. sylvestris*-cloned DNA was performed using SeaView v.4 software [87]. The percent of amino acid identity was also determined using the EMBOSS Needle online tool.

### 5.6. qRT-PCR

Isolation of total RNA was performed using an extraction CTAB buffer (pH 4.8–5.0): 100 mM Tris–HCl (pH 8.0), 25 mM EDTA, 2M NaCl, 2% CTAB, 2% PVP-40; 2% mercaptethanol was added to the mixture before use. Separation of the aqueous and organic phases was done using a mixture of chloroform-isoamyl alcohol (24:1). RNA was precipitated using 25 mM LiCl, then re-precipitation was carried out using an extraction SDS buffer: 1M NaCl, 0.5% SDS, 10 mM Tris–HCl (pH 8.0), 1 mM EDTA [88]. RNA was re-precipitated with absolute isopropanol. The quality and quantity of the isolated RNA were checked spectrophotometrically (absorbance microplate and cuvette reader SPECTROstar NANO, “BMG Labtech”, Ortenberg Germany) and by electrophoresis in 1% agarose gel (Sub-Cell GT Agarose Gel electrophoresis systems, Bio-Rad, USA). Before reverse transcription (RT), total RNA was additionally treated with DNase and inhibitor of RNase (Syntol, Moscow, Russia) (incubation for 1 h at 37 °C). DNase inactivation was carried out during the RT reaction by heating the reaction mixture (70 °C) for 10 min before the reaction start.

Specific primers (Syntol, Moscow, Russia) for amplification of the studied genes were designed using the software Beacon Designer 8.21 (PREMIER Biosoft) (Table 3). As a reference gene for normalization of quantitative PCR data, we assume to use the stably expressed *GAPDH* gene.

We performed amplification of the samples using a real-time detection system CFX96 (“BioRad”, Hercules, CA, USA) and the amplification kit with an intercalating dye SYBR Green (Evrogen, Moscow, Russia). The specificity of amplification products was checked by melting the PCR fragments and by using 8% acrylamide gel electrophoresis. Gel electrophoresis analysis indicates that all of the amplicons were of the expected length. The relative quantity of gene transcripts (RQ) was calculated from the formula:RQ = E^−ΔCt^,(1)
where ΔCt is the difference of the threshold cycle values for the reference and target genes, and E—effectiveness of PCR.

Amplification efficiency was determined individually in each reaction based on raw amplification fluorescence data using LinRegPCR software [89]. We calculated relative expression using the average reaction efficiency values for each pair of primers.

The level of expression of specific genes was expressed in relative units (arbitrary units).

The number of biological replicates (i.e., model trees) was fivefold for 30-year-old and 170–180-year-old groups, sevenfold for 70–80-year-old groups, and eightfold for the 40-year-old. The number of technical replicates ranged from one to two for each model tree.

### 5.7. Statistical Data Processing

The results were statistically processed with PAST (version 4.0). Before starting the statistical analysis, we initially tested raw data for normality using the Shapiro–Wilk test. The Mann–Whitney U-test estimated the significance of differences between variants. Different letters show a significant difference at *p* < 0.05. All data in the diagrams appear as mean ± SE, where SE is the standard error.

The research was carried out using the equipment of the Core Facility of the Karelian Research Centre of the Russian Academy of Sciences.

## Figures and Tables

**Figure 1 plants-12-00835-f001:**
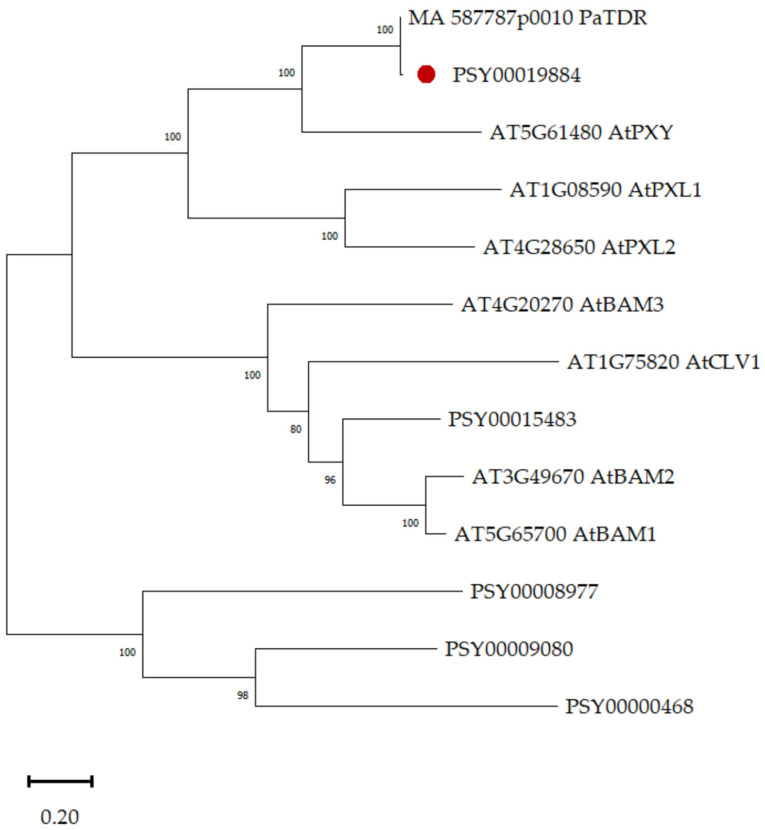
Phylogenetic relationships built based on a comparative evolutionary analysis of the amino acid sequences of potential Scotch pine (PSY) proteins and some known kinases of the LRR-RLK family of subgroup XI of *Picea abies* and *Arabidopsis thaliana*. Phylogenetic analysis by maximum likelihood (ML) method, based on LG with Freqs. (+F) model with a discrete gamma distribution (1 G, 5 categories) as the best model for gene phylogeny reconstruction. Branch length is estimated at genetic distance (number of substitutions per site). Significant bootstrap values (percentage of trees in which associated taxa are clustered together) for 1000 replicates are shown at the branch base. An access code in the Phytozome and Congenie databases is next to the protein names. The red dot is the protein encoded by the *PsPXY*, for which expression was analyzed.

**Figure 2 plants-12-00835-f002:**
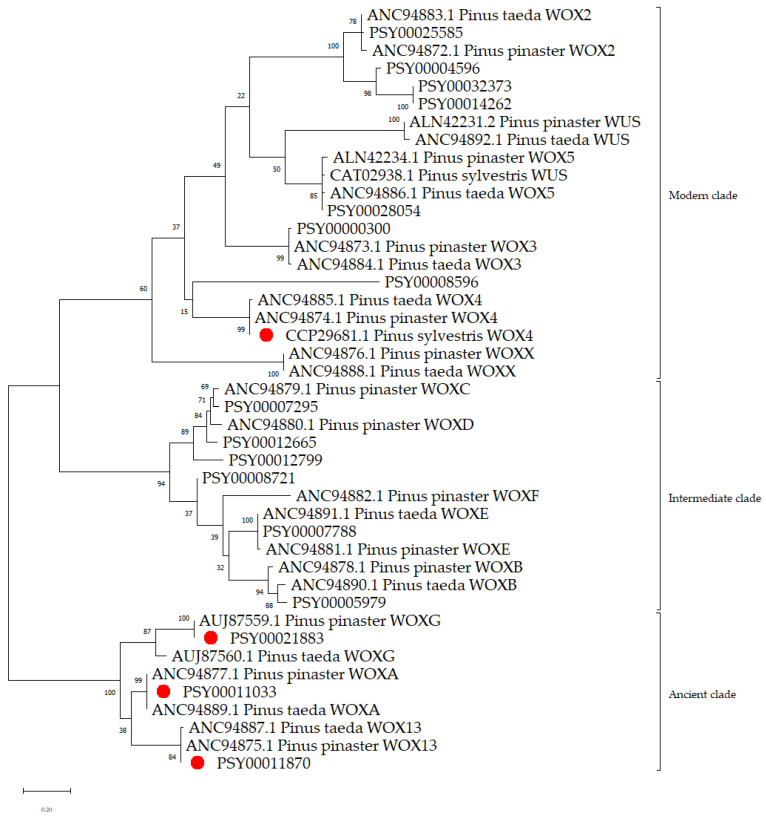
Phylogenetic relationships built based on a comparative evolutionary analysis of the amino acid sequences of potential Scotch pine WUS/WOX proteins (PSY) and known proteins of the WUS/WOX family of other *Pinus* species. Phylogenetic analysis by maximum likelihood (ML) method, based on Jones–Taylor–Thornton (JTT) model with a discrete gamma distribution (1 G, 5 categories) as the best model for gene phylogeny reconstruction. Branch length is estimated at genetic distance (number of substitutions per site). Significant bootstrap values (percentage of trees in which associated taxa are clustered together) for 1000 replicates are shown at the branch base. An access code in the NCBI database is next to the protein names. The red dots are the proteins encoded by the *PsWOXs*, which were analyzed.

**Figure 3 plants-12-00835-f003:**
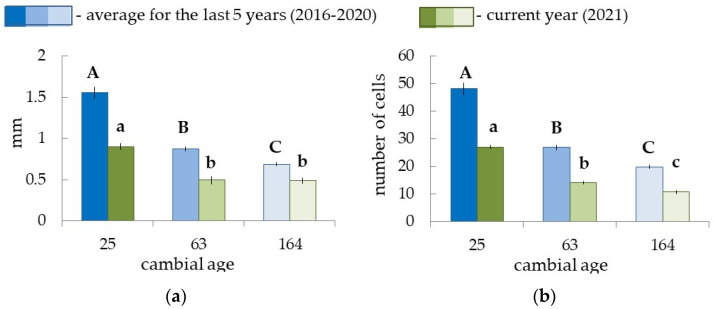
The width of annual xylem increment (**a**) and number of cells in the radial row of the xylem (**b**) in pine plants of different ages. The blue color—the average meanings for the last 5 years, 2016–2020 (samples were collected in May 2021) and the green color—the meanings of current year xylem (samples were collected on 21–22 June 2021). Different letters indicate significant differences at *p*-value ˂ 0.05. Capital letters show significant differences between average meanings for the last 5 years, lower ones—for the current year.

**Figure 4 plants-12-00835-f004:**
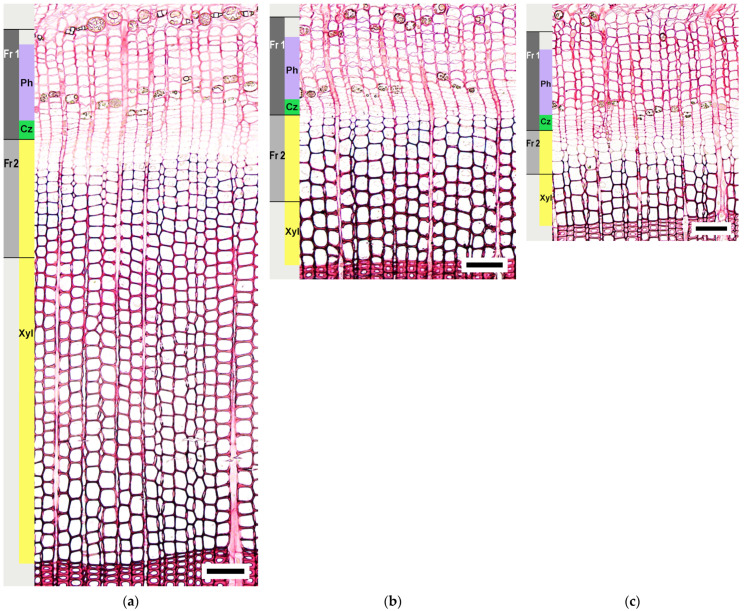
Transverse trunk tissue sections include conductive phloem (Ph), cambial zone (Cz), and current year xylem (Xyl) in 30-year-old (**a**), 70–80-year-old (**b**), and 170–180-year-old (**c**) pine trees. Scale bar = 100 µm. Samples collected on 21–22 June 2021. Tissues sampled for the identification of gene expression level indicated as Fr1 (Fraction1) and Fr2 (Fraction2).

**Figure 5 plants-12-00835-f005:**
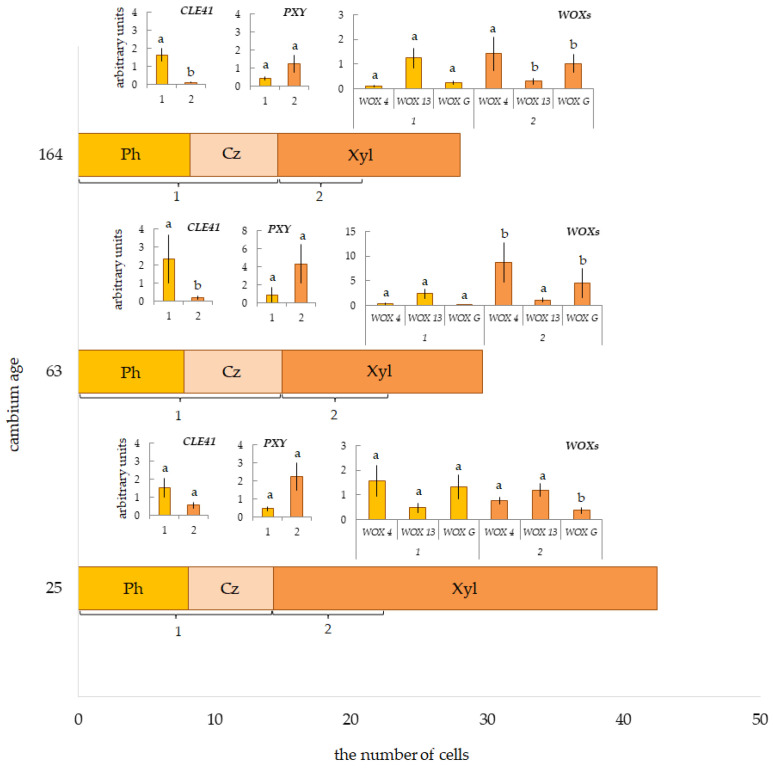
The number of cells in the radial row of conducting phloem (Ph), cambial zone (Cz), and current year xylem (Xyl) in pine plants of different ages. Relative expression of genes *CLE41/PXY/WOXs* in tissues, including the cambial zone, differentiating phloem and mature phloem (Fraction 1 (1)), and differentiating xylem (Fraction 2 (2)). Samples collected on 21–22 June 2021. Different letters show differences only between Fraction 1 and Fraction 2.

**Figure 6 plants-12-00835-f006:**
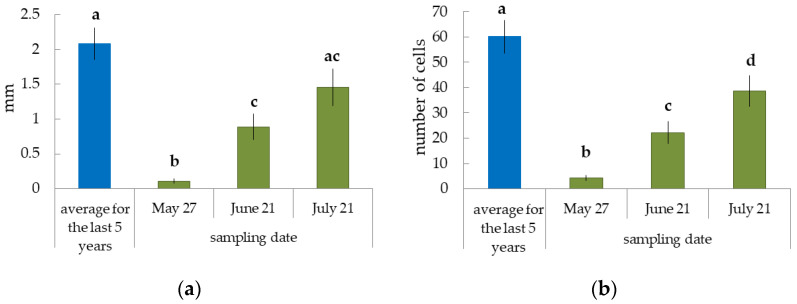
The width of annual xylem increment (**a**) and number of cells in the radial raw of the xylem (**b**) in 40-year-old pine trees. The blue color—the average meanings for the last 5 years, 2017–2021 (samples were collected in May 2022) and the green color—the meanings of current year xylem (Samples collected in May–July 2022). Different letters show significant differences at *p*-value ˂ 0.05.

**Figure 7 plants-12-00835-f007:**
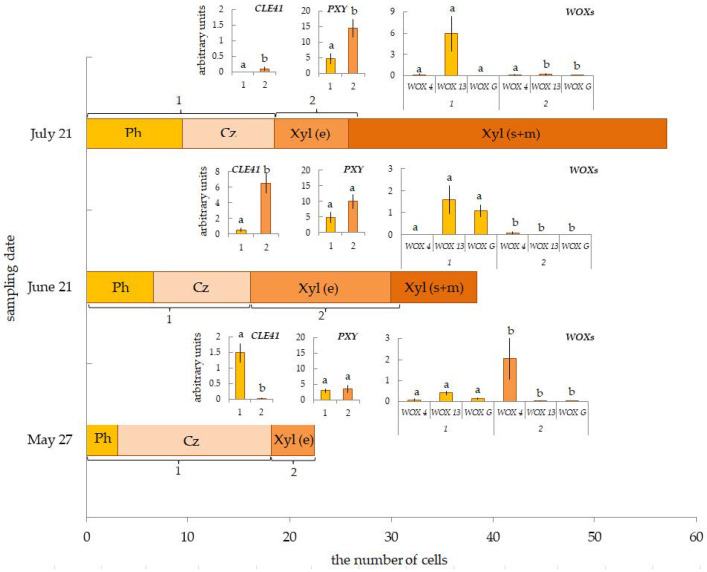
The number of cells in the radial row of conducting phloem (Ph), cambial zone (Cz), expanding cells (Xyl (e)) and cells that are forming secondary cell wall, and mature cells (Xyl (s+m)) of current year xylem in 40-year-old pine plants. Relative expression of genes *CLE41/PXYWOXs* in tissues, including the cambial zone, differentiating phloem and mature phloem (Fraction 1 (1)), and differentiating xylem (Fraction 2 (2)). Samples collected on 27 May, 21 June, and 21 July 2022. Different letters show differences only between Fraction 1 and Fraction 2.

**Figure 8 plants-12-00835-f008:**
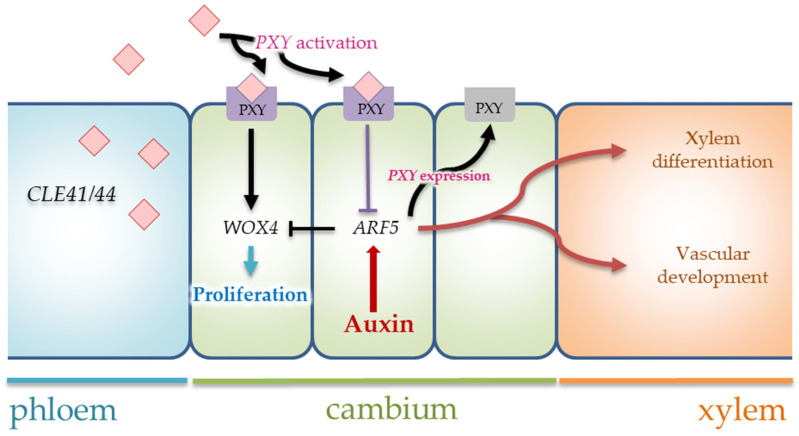
Model showing interactions between TDIF(CLE41)/TDR(PXY) signaling and auxin signaling for proliferation/differentiation cambium derivatives. The explanations are in the text.

**Table 1 plants-12-00835-t001:** Features of the P. sylvestris subgroup XI receptor kinase-like (RLK) family members.

Gene Name	Gene ID	Number of Amino Acid Residues in Peptide	Location of Transmembrane Helices	Location of Protein Kinase Domain	Subcellular Localization (Likelihood)
	*PSY00000468*	1145	738–758	803–1115	Cell membrane (0.7873)
	*PSY00008977*	1121	761–781	830–1121	Cell membrane (0.8066)
	*PSY00009080*	790	435–455	502–782	Cell membrane (0.8222)
	*PSY00015483*	1003	638–657	696–988	Cell membrane (0.7722)
*PsPXY*	*PSY00019884*	515	135–155	204–500	Cell membrane (0.8176)

**Table 2 plants-12-00835-t002:** Features of the *P. sylvestris* WUSCHEL-related homeobox (WOX) family members.

Gene Name	Gene ID	Number of Amino Acid Residues in Peptide	Location of Homeobox Domain	Subcellular Localization (Likelihood)
	*PSY00000300*	147	7–146	Nucleus (0.9137)
	*PSY00004596*	217	41–100	Nucleus (0.9163)
	*PSY00005979*	260	42–98	Nucleus (0.9250)
	*PSY00007295*	273	55–114	Nucleus (0.9078)
	*PSY00007788*	257	53–113	Nucleus (0.9104)
	*PSY00008596*	194	32–90	Nucleus (0.8805)
	*PSY00008721*	253	53–94	Nucleus (0.9227)
*PsWOXA*	*PSY00011033*	418	140–198	Nucleus (0.8991)
*PsWOX13*	*PSY00011870*	356	85–143	Nucleus (0.9085)
	*PSY00012665*	274	55–116	Nucleus (0.9178)
	*PSY00012799*	274	55–114	Nucleus (0.9088)
	*PSY00014262*	194	29–88	Nucleus (0.9282)
*PsWOXG*	*PSY00021883*	294	100–158	Nucleus (0.9324)
	*PSY00025585*	186	10–64	Nucleus (0.9270)
	*PSY00028054*	284	49–108	Nucleus (0.8816)
	*PSY00032373*	178	25–84	Nucleus (0.9336)
*PsWOX4*	*CCP29681.1* (NCBI)	478	130–189	Nucleus (0.9157)

**Table 3 plants-12-00835-t003:** List of primers for the RT-PCR reaction.

Gene Name	ID of Sequences Used for the Primer Design	Forward Primer (5′→3′)	Reverse Primer (5′→3′)	Ta, °C
*PsGAPDH*	*PSY00009485*	GGACAGTGGAAGCATCAT	AACCGAATACAGCAACAGA	54
*PsCLE41/44*	*PTA00040742* *PPI00060734*	GTATGGCGGATGGTTTTG	ATTACTAATTGGATTTGGACCG	55
*PsPXY*	*PSY00019884*	GTTGCCTTCCATTACAGA	GGTCCGTTAAGATGATTGA	60
*PsWOX4*	*CCP29681.1*	ACTATACTAACGAAGAAGA	TAATACTGAGTTGTCCAT	53
*PsWOX13*	*PSY00011870*	TGTGTCTGGTCAAGGATT	TCTCTAAGATATGAAGTTGTGTT	59
*PsWOXG*	*PSY00021883*	TGGATAATAGCCTTGACT	CACTGTTGAGTATCATCTT	55

## Data Availability

All data included in the main text and Appendix A.

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
