# Peer review of "Identification and Expression Profile of CLE41/44-PXY-WOX Genes in Adult Trees Pinus sylvestris L. Trunk Tissues during Cambial Activity"

_plants, 2023, doi:10.3390/plants12040835_

Round 1

Reviewer 1 Report (New Reviewer)

      1.   English is not smooth with unusual vocabulary. A thorough language editing could

            make the manuscript easy to read.

       2. Lot of grammatical errors in the manuscript. So, here is a need of

           deep revision of every line.

3. Write down the botanical name of plants in the entire manuscript and italicize them.

4. Repetitions should be avoided in the whole manuscript.

5. Abstract is too lengthy and descriptive. It should be clear and concised. 

6. Introduction needs to be modified. Wox genes have been identified and characterized in numerous plant species, for example https://www.sciencedirect.com/science/article/pii/S2352407318300118, https://www.sciencedirect.com/science/article/pii/S2405844020326050 etc. However, authors have discussed mostly about Arabidopsis.  

6. Line no. 81 to 85 should be re-written.

7. Line no. 107-108 should be rephrased.

Line 130, 2.1. CLE41/44 and PXY gene identification in the Scots pine genome. CLE41/44 gene sequencing ????? Author should carefully check the Ms before submission.

8. Table 1. Why location of TMs not predicted in a few proteins, author should try different tools.

9. Section 2.3 – 2.7 are not seems to be correlated with each other, why they have been done, sgould be clearly mentioned. Title should also be changed accordingly.

10. The discussion section is too bulky. The key findings should be discussed instead of

     detailing the results.

11. Conclusion should not be repeat of abstract.

Author Response

We are grateful to you for your detailed analysis of our work, recommendations and comments, to which we present our responses below.

Comments and Suggestions for Authors

  1. English is not smooth with unusual vocabulary. A thorough language editing could make the manuscript easy to read.

Our answer. We have done language editing.

  1. Lot of grammatical errors in the manuscript. So, here is a need of deep revision of every line.

Our answer. We have done deep revision.

  1. Write down the botanical name of plants in the entire manuscript and italicize them.

Our answer. We have changed the text of the manuscript.

  1. Repetitions should be avoided in the whole manuscript.

Our answer. We've made changes.

  1. Abstract is too lengthy and descriptive. It should be clear and concised.

Our answer: We've changed to the Abstract.

  1. Introduction needs to be modified. Wox genes have been identified and characterized in numerous plant species, for example https://www.sciencedirect.com/science/article/pii/S2352407318300118, https://www.sciencedirect.com/science/article/pii/S2405844020326050 etc. However, authors have discussed mostly about Arabidopsis.

Our answer: The experimental models widely used to study the molecular-genetic mechanisms of xylogenesis regulation are Zinnia elegans L. cell culture, where in a medium with auxin and cytokinin mesophilic cells transdifferentiate into tracheal elements, and Arabidopsis thaliana, in whose hypocotyl the types of secondary xylem vascular cells are similar to the types found in poplar, but smaller and lacking ray parenchyma cells. Module have been widely described on the example of Arabidopsis thaliana functions of CLE-PXY-WOX signaling. Populus trichocarpa is considered a model species for studying the molecular genetic aspects of regulating xylogenesis among woody plants [for example, Etchells et al., 2015 doi:10.1016/j.cub.2015.02.023]. Usually, studies are carried out on seedlings in which one or two rings of juvenile wood are formed. Published transcriptomes of woody plants contain information on the distribution of CLE41-PXY-WOX4 gene expression in trunk tissues of adult plants [for example, Alonso-Serra et al., 2019 doi:10.1111/nph.15725], but such works are unitary and do not contain information about the seasonal dynamics. The information about the function of ancient WOXs genes in conifers is not available. Expression patterns of ancient WOX genes are present in most tissues, as shown in three-week-old seedlings and somatic embryos of Pinus pinaster or Pinus taeda [for example Alvarez et al., 2018 doi:10.1016/j.plaphy.2017.12.031]. We have not found functional studies of these genes in gymnosperm species during the cambial growth period.

  1. Line no. 81 to 85 should be re-written.

Our answer: We've made changes.

  1. Line no. 107-108 should be rephrased.

Line 130, 2.1. CLE41/44 and PXY gene identification in the Scots pine genome. CLE41/44 gene sequencing ????? Author should carefully check the Ms before submission.

Our answer: We've made changes.

  1. Table 1. Why location of TMs not predicted in a few proteins, author should try different tools.

Our answer: We thank the Reviewer for the valuable comment. We used a new tool for better TM prediction. We added the information to the Table 1 in the results section, as well as to the Material and Methods section.

  1. Section 2.3 – 2.7 are not seems to be correlated with each other, why they have been done, sgould be clearly mentioned. Title should also be changed accordingly.

Our answer: We've made some changes.

Section 2.3. We changed name to «Characteristics of cambial activity in pine trees of different ages growing in lingonberry pine forests». Since cambial activity is highly changeable and can be varied throughout the plant’s life cycle, we determined how cambial activity differed in the objects of study – trees of different ages.

Section 2.4. We changed name to «Characteristics of xylogenesis, phloem formation in the current year, and cambial zone in pine trees of different ages». We characterized the conducting tissues and the cambial zone in trees of different ages to determine the stage of seasonal development at the sampling period for molecular genetic analysis.

Section 2.5. Expression of the genes encoding CLE41/44-PXY-WOX –signaling module in trunk tissues of Scots pine of different ages. We studied the expression patterns of the CLE41/44-PXY-WOX genes, which control the proliferation of cambial cells in the trunk tissues of pine trees of different ages. Since, during the sampling period (June 21-22, 2021), we could not see differences in CLE41/PXY expression patterns in trees of different ages, in 2022 we studied changes in CLE41/PXY/WOXs gene expression in trunk tissues of 40-year-old pines on different stages of cambial growth.

Section 2.6. Characteristics of cambial activity in the 40-year-old pine trees at different stages of cambial growth. We presented data on the anatomical characteristics of conducting tissues and the cambial zone, as well as the stage of seasonal development on each date of sampling for molecular genetic studies in 40-year-old pine trees.

Section 2.7. We changed name to «Expression of the genes encoding CLE41/44-PXY-WOX signaling module in in the 40-year-old pine trees at different stages of cambial growth».

  1. The discussion section is too bulky. The key findings should be discussed instead of detailing the results.

Our answer: We have cut the discussion section.

  1. Conclusion should not be repeat of abstract.

Our answer: We've made changes.

Sincerely,

Natalia A. Galibina, Yulia L. Moshchenskaya, Tatiana V. Tarelkina, Kseniya M. Nikerova, Maxim A. Korzhenevskii, Aleksandra A. Serkova, Nikita V. Afoshin, Ludmila I. Semenova, Diana S. Ivanova, Elena N. Guljaeva and Olga V. Chirva

Reviewer 2 Report (New Reviewer)

The manuscript submitted by Galibina et al. describes the expression profile of the WOX genes in Scots pine trees of different ages for cambial growth. The authors concluded that WOX genes expression patterns and signaling regulate cambial growth. The main research findings of this paper will be important for the full understanding of molecular mechanisms of cambial growth. However, a few modifications to the following points would make it clearer to the readers.

Figure 2 should clearly show the three characteristic clades.

I would like the authors to represent where Fraction 1 and 2 are located in Fig. 4, 5 and 7.

This manuscript describes differences in WOX gene expression between cambium ages (Fig. 5) and growth date (Fig. 7). If the authors would like to show differences between cambium ages and growth date, these data should be combined into one graph.

The PsWOX gene, whose expression timing and location are tightly regulated, plays an important role in cell proliferation and the formation of the cambial zone. These relationships are a bit complex. It would be useful to include a diagram of hypothetical working model.

Author Response

We are grateful to you for your detailed analysis of our work, recommendations and comments, to which we present our responses below.

Comments and Suggestions for Authors

Figure 2 should clearly show the three characteristic clades.

Our answer. We've changed Figure 2.

I would like the authors to represent where Fraction 1 and 2 are located in Fig. 4, 5 and 7.

Our answer. We showed localization of Fraction 1 and 2 in Figures 4, 5, and 7.

This manuscript describes differences in WOX gene expression between cambium ages (Fig. 5) and growth date (Fig. 7). If the authors would like to show differences between cambium ages and growth date, these data should be combined into one graph.

Our answer. Thanks for the suggestion. You are right, it would be interesting to consider together the differences in gene expression between cambium ages and cambium growth stage. We plan to conduct a similar study in our future work.

In this work, the expression of CLE41/44-PXY-WOXs genes was studied depending on the stage of ontogeny (the first group of objects) and depending on the stage of cambial growth (the second group of objects). The first group of objects included lingonberry pine forests of different ages (30, 80, and 180-year-old stands) in the middle taiga subzone. We sampled stem tissues during the period (June 22-23, 2021), when late phloem and early xylem were formed in the stem. The second group of objects, included 40-year-old pine trees, were selected growing in the forest seed orchard. We carried stem tissue sampling out in different phases of cambial growth (on 2022 May 27, June 21 and July 21). Since (1) the objects differ significantly from each other, and (2) the weather in 2021 and 2022 were different, we believe it would be incorrect to combine data for these two groups of objects.

The PsWOX gene, whose expression timing and location are tightly regulated, plays an important role in cell proliferation and the formation of the cambial zone. These relationships are a bit complex. It would be useful to include a diagram of hypothetical working model.

Our answer. We added Figure 8 «Model showing interactions between TDIF(CLE41)/TDR(PXY) signaling and auxin signaling for proliferation/differentiation cambium derivatives».

Sincerely,

Natalia A. Galibina, Yulia L. Moshchenskaya, Tatiana V. Tarelkina, Kseniya M. Nikerova, Maxim A. Korzhenevskii, Aleksandra A. Serkova, Nikita V. Afoshin, Ludmila I. Semenova, Diana S. Ivanova, Elena N. Guljaeva and Olga V. Chirva

Round 2

Reviewer 1 Report (New Reviewer)

This study seems to be not enough to be considered for the conclusions drawn. Authors have did only RT PCR, not even localization study has been done. 

Therefore, authors should either do the inclusive characterization as proposed in conclusion or they may change the title and conclusion for gene family analysis with Minimal functional data, with comparative discussion with the wox genes in other plants. 

Author Response

Comments and Suggestions for Authors

This study seems to be not enough to be considered for the conclusions drawn. Authors have did only RT PCR, not even localization study has been done. Therefore, authors should either do the inclusive characterization as proposed in conclusion or they may change the title and conclusion for gene family analysis with Minimal functional data, with comparative discussion with the wox genes in other plants.

Our answer.

We are grateful to you for your recommendations and comments, to which we present our responses below.

We changed the title according to the reviewer's recommendation to « Identification and expression profile of CLE41/44-PXY-WOX genes in adult trees Pinus sylvestris L. trunk tissues during cambial activity».

We have also changed the conclusion.

The study of the regulatory mechanisms of cambial growth has been carried out for several decades. Scientists have accumulated a large amount of data on anatomical and biochemical changes in the cells of the cambial zone during the period of activity (Larson 1994). Also, many works have been devoted to the hormonal regulation of this process (Savidge 2001; Sorce et al. 2013; Buttò et al. 2020). Then it was found that peptide hormones and transcription factors associated with them are also involved in regulating cambial activity (Etchells et al. 2016). Elements of the CLE41/44-PXY-WOX signaling module have been identified in several angiosperm and gymnosperm species of woody plants; however, the expression of the corresponding genes has been studied mainly in young plants less than 10 years old (Gambino et al. 2011; Hedman et al. 2013; Etchells et al. 2015; Kucukoglu et al. 2017; Alvarez et al. 2018). There are only a few single works, during which the features of CLE41/44-PXY-WOXs gene expression were studied on plants older than 10 years (Galibina et al. 2022). Thus, there is a gap in our knowledge about the activity of the cambium of adult trees on the anatomical and biochemical level, on the one hand, and regulating this process at the molecular level.

In our work, we focused on the study of CLE41/44-PXY-WOXs gene expression in the cambial zone of the trunk of adult trees aged from 30 to 180 years, growing under natural conditions without anthropogenic influence. We also performed a microscopic analysis of developing tissues to accurately characterize the stage of cambial activity. We know that there is a very limited range of methods that can be applied to objects of this age, considering the conservation status of the territory where the trees grew. We believe our data are important for understanding the molecular mechanisms of regulation of cambial growth in mature trees forming natural forests, as well as at different stages during the growing season. Our data may also help other researchers in selecting objects and timing of sampling to study the regulation of cambial growth in mature trees.

Sincerely,

Natalia A. Galibina, Yulia L. Moshchenskaya, Tatiana V. Tarelkina, Kseniya M. Nikerova, Maxim A. Korzhenevskii, Aleksandra A. Serkova, Nikita V. Afoshin, Ludmila I. Semenova, Diana S. Ivanova, Elena N. Guljaeva and Olga V. Chirva

Round 3

Reviewer 1 Report (New Reviewer)

Ms have been improved and focused as compared to earlier version. But authors should mention about the wox genes status in other plants also in the introduction and discussion section as suggested during the first round of review. 

Author Response

Comments and Suggestions for Authors

Ms have been improved and focused as compared to earlier version. But authors should mention about the wox genes status in other plants also in the introduction and discussion section as suggested during the first round of review.

Our answer.

We are grateful to you for your recommendations and comments.

And we have made necessary changes to the manuscript text.

Sincerely,

Natalia A. Galibina, Yulia L. Moshchenskaya, Tatiana V. Tarelkina, Kseniya M. Nikerova, Maxim A. Korzhenevskii, Aleksandra A. Serkova, Nikita V. Afoshin, Ludmila I. Semenova, Diana S. Ivanova, Elena N. Guljaeva and Olga V. Chirva

This manuscript is a resubmission of an earlier submission. The following is a list of the peer review reports and author responses from that submission.

Round 1

Reviewer 1 Report

The authors have modified the text and included new results. In the new version the contents are better organized, specially in the result and discussion sections (which in this version are separate sections) and some of the ideas are more clearly stated and organized. The new data consist on the tracking of changes in the number of cells in the radial row and the changes in the expression of CLE41/PXY/WOXs genes in trunk tissues at different stages of cambial growth in 40-year-old pine. However, despite the effort still remain important weaknesses in the manuscript:

The English language have some improvement, but still needs a major review to improve quality to publication standards. Some words appear in Cyrillic characters.

The main weakness of the manuscript is the highly speculative conclusions and weakly supported by the data presented, since the authors attempt to establish functional conclusions for genes potentially involved in the control of the development of the secondary vascular system based exclusively on expression profiles. As the authors acknowledge in response to the reviewer's comments, the conclusions are merely hypotheses that they plan to test in the future. Particularly striking are the conclusions about the genes of the ancient clade of the WOX genes, WOX13 and WOXG. For the former, the authors conclude, and even highlight in the abstract, that “Based on our results and literature data, there is an assumption about the role of PsWOX13 in the control of maternal phloem cell proliferation. Confirmation of this hypothesis requires further research". By reading the discussion about the results obtained for this gene and some references to the literature that the authors include in the discussion this proposal is highly speculative. In the case of the WOXG gene the authors acknowledge that they cannot provide an explanation for their data, which shows the excessively preliminary state of the work.

Other aspects to consider:

An alignment and comparative analysis of the amino acid sequences of CLE41 peptides from P. pinaster and P. taeda with the  sequence derivated from P. sylvestris cloned DNA should be included, at least as a supplementary file. Also, has this sequence been deposited in a Public Gene Database? Accession number should be provided.

Figure 6. Figure legends indicates “Different letters indicate significant differences at p-value Ë‚ 0.05. 338”, but there are not such letters in the figure.

Figure 5 and Figure 7. Statistical significance of differences are not indicated in relative gene expression charts.

Materials and Methods section, 5.5. Sequencing. The first paragraph refers to DNA being isolated from xylem samples, but the description of the reaction components for PCR specifies that cDNA was added. Correct/clarify.

Reviewer 2 Report

The authors have submitted the revised version of their manuscript studying the spatial and temporal expression of CLE41/44-PXY-WOX genes in trunks of Scots pines. Especially addition of the temporal study and the expression analyses of ancient clade WOXA and WOXG that are lacking orthologs in angiosperms makes the revised version interesting for wider audience. However, the discussion concerning the expression of WOXG is very superficial and should be improved. In addition, there was only one reference gene in qPCR analyses. This is not adequate. MIQE guidelines suggest the use of at least two reference genes. Please also remove different language than English from the legend of Figure 2.